# OpenReview forum: "Reproducibility Study of "Cooperate or Collapse: Emergence of Sustainable Cooperation in a Society of LLM Agents""
_TMLR — Rejected by TMLR_

### Review · Reviewer_yAiJ · 2025-02-20

**Summary Of Contributions:**

The study replicates key experiments from Piatti et al. (2024) for sustainable cooperation of LLMs agents in resource-sharing scenarios, validating the claims in Piatti et al. (2024) that all but the most powerful LLM agents fail to achieve a sustainable equilibrium, and that agents instructed to leverage “Universalization”-based reasoning are able to achieve better sustainability.

The paper additionally provides the following extensions or new settings:
- New model evaluations: additional models—such as DeepSeek-V3, GPT-4o-mini, and smaller variants from the Qwen family—are tested to determine if cooperative behavior generalizes across different architectures and model sizes.
- Language Variation: the impact of language is explored by translating the agent instructions into Japanese. The authors frame this experiment as testing whether cultural-linguistic nuances (e.g., collectivist tendencies) affect cooperation.
- Inverse Environment Scenario: a novel “trash” scenario is introduced in which the shared resource is harmful (mathematically equivalent to a positive scenario). Here, agents must work together to eliminate a detrimental resource rather than to harvest a beneficial one.
- Heterogeneous Multi-Agent (MultiGov) Scenario: the framework is extended to allow different agents (using different LLMs) to interact, instead of using the same LLMs for all agents.

**Audience:**

Yes

**Broader Impact Concerns:**

No immediate broader impact concerns.

**Claims And Evidence:**

No

**Requested Changes:**

- Expand Experimental Scope and Robustness: See the above section. While this is not strictly necessary, and the authors' claim for budget constraints are reasonable, more experiments will provide more robust credence to the results, especially for the novel observations like the inverse setting.
- Revise and Strengthen the Language Setting Experiment: See the above Section
- Clarify Experimental Limitations and Future Directions:
  - Provide more clarity of the constraints (computational and cost-related) that led to focusing solely on the fishery scenario and reducing the number of runs, and addressing potential arguments to the limitations to scope and generalizability.
- Better result presentation: the arrangement of table 6 is slightly confusing. It will help to present the results by highlighting how the survival rate changes particularly for the 3 scenarios that are changing sequentially (4 DeepSeek-V3, 1 gpt4omini; 3 DeepSeek-V3, 2 gpt4omini; 1 DeepSeek-V3, 4 gpt4omini). Also, this seems helpful to include the missing scenario (2 DeepSeek-V3, 3 gpt4omini) so a full range is visible. The other experiment (4 gpt4o 1 gpt4omini) should ideally not be in the middle of this sequence, obstructing this progression.
- Are there any example reasonings or dialogues of LLMs that will be helpful for illustrating the results? The analysis and results could be augmented by example behaviors highlighting how, e.g. high-performing LLMs like deepseek v3 guides gpt4omini to better survival.

**Strengths And Weaknesses:**

Strengths
- Well-Presented Work
   - The paper is clearly written and structured, making its methods and findings accessible.
- Novel Framing Observation
   - The inverse experiment scenario yields interesting insights into framing effects.

Weaknesses:
- Limited experimental scope:
  - The experiments are confined to the *fishery* scenario only, excluding the *pasture* and *pollution* scenarios featured in the original GovSim paper.
  - The focus is solely on the default and universalization setups, and each experimental setting is run only three times (reduced from five as in Piatti et al. (2024)), which may undercut the robustness of the insights. This is particularly concerning given that the original study reported marked differences between scenarios, noting that "LLMs performed better in the fishery scenario than in the pasture and pollution scenarios."
- Questionable Motivation for Language Setting:
  - The authors justify including Japanese text based on the hypothesis that language models trained on Japanese might exhibit behaviors aligned with collaboration and mutual benefit due to collectivist cultural influences.
  - However, the experiments were conducted on GPT-4o and DeepSeek V3, for which available evidence suggests that Japanese data constitute only a very small fraction of their training sets (e.g., GPT-3 contained less than 0.17% Japanese text). Similarly, DeepSeek V3, produced by a Chinese firm, likely relies predominantly on English and Chinese data.
  - As such, the current experimental setup does not convincingly support the claim that training on Japanese fosters could lead to potentially enhanced cooperation. A more compelling argument would require either using models specifically tuned for Japanese or by explicitly comparing models trained predominantly on languages with strong collectivist influences (like Chinese, as with DeepSeek), to clearly isolate the impact of language on collaborative behavior.

---

> ### Author Response · Authors · 2025-04-18
>
> We would like to thank the reviewer for the careful reading of our manuscript and for the positive and helpful comments. We have replied to all of them, as described in detail below, and we believe that they have led to an improvement of the paper. A highlighted version of the manuscript with the corrections in red was uploaded.
>
> ## Expand Experimental Scope and Robustness
>
> We acknowledge the reviewer’s point that additional experiments would enhance the robustness of our results. We agree that this is a limitation and have added new experimental results to this section. The new experiments include results averaged over five runs instead of three, still falling within the error margin of the original results, increasing our certainty about their robustness. Initially, we were constrained to three runs per experiment since our computational budget was (and still is) limited.
>
> We now have the exact number of runs the original authors had in their results.
> Furthermore, we have added explanations regarding this limitation in the main text, addressing the following:
> - Our study prioritizes reproducibility. While three runs are a limitation, it aligns with the original paper's margin of error and is sufficient for validation.
> - For novel experiments, we used five runs, such as the original paper's authors, to improve robustness and expand to other scenarios.
>
> ## Revise and Strengthen the Language Setting Experiment
>
> We appreciate the reviewer’s insightful comment about the motivation for including Japanese text and recognize the limitation of our study. As noted, the models we used (GPT-4o and DeepSeek-V3) are not specially trained on Japanese, which limits the strength of the original hypothesis that training in Japanese might enhance cooperation.
>
> However, our justification stems from findings from prior work (Dudy2024), which found that LLMs can still display culturally influenced behavior through language. That study showed differences in how LLMs responded to emotional scenarios in East Asia, suggesting that language alone may shape model behavior in ways that reflect cultural norms. This supports our hypothesis that language itself can influence cooperation.
>
> While we agree that models explicitly trained in Japanese would be ideal, we were limited by resource constraints and access. We have now clarified this motivation and limitation in the Japanese translation section.
>
>
>
> ## Clarify Experimental Limitations and Future Directions
> ### Provide more clarity of the constraints
> - Our computational constraints are now clearly detailed in Section 3.2, along with the adjustments described in points 1 and 2. To address concerns about generalizability, we have added a new paragraph in Section 3.2 and expanded on these trade-offs in the Broader Impact and Ethical Considerations section, where we explain how resource limitations influenced our scenario selection and outline plans to broaden the evaluation scope in future work as resources allow.
>
> ### Better result presentation
>
> - We agree with the reviewer’s suggestion regarding the presentation of the results in Table 6. To improve clarity, we have reorganized the table to highlight how the survival rate changes across the sequentially changing scenarios and have included the missing scenario (2 DeepSeek-V3, 3 GPT-4o-Mini) to provide a complete range of results. We also moved the experiment (4 GPT-4o, 1 GPT-4o-Mini) out of the middle of this sequence to avoid obstructing the progression.
>
> ### Are there any example reasonings or dialogues of LLMs that will be helpful for illustrating the results?
>
> - We appreciate the reviewer’s suggestion and agree that providing an example of such communication would significantly enhance the clarity of our results. Therefore, we have included a communication example between DeepSeek-V3 and GPT-4o-Mini in the main content, now under Fig. 2, to better illustrate the interaction and support our findings.

---

### Review · Reviewer_aoB9 · 2025-03-09

**Summary Of Contributions:**

This paper validates the GovSim framework, confirming that large LLMs like GPT-4-turbo achieve sustainable cooperation in resource-sharing scenarios, while smaller models need the universalization principle. Key extensions include: model generalization, showing varied cooperative behavior across architectures with DeepSeek-V3 matching GPT-4-turbo; cross-lingual analysis, where Japanese instructions did not alter behavior, challenging assumptions about cultural collectivism; inverse environment scenarios, where agents eliminated harmful resources, highlighting the impact of scenario framing on cooperation; and heterogeneous multi-agent systems, where high-performing models improved low-performing agents' behavior through communication, suggesting scalable cooperative systems. These findings underscore LLMs' adaptability in cooperative tasks and offer insights for designing efficient multi-agent systems.

**Audience:**

Yes

**Broader Impact Concerns:**

- The ability of high-performing models to influence others could be misused in adversarial settings (e.g., spreading misinformation). A broader impact statement should address this.
- The Japanese experiment’s design (e.g., non-native translation) raises questions about cultural representation in LLM evaluations. A statement on limitations and ethical data practices is needed.

**Claims And Evidence:**

Yes

**Requested Changes:**

- Supplement the impact of different languages on agents: besides the type of the text itself, does the related background description also affect the agent's behavior? For instance, if the event occurs at a specific fishing ground in a particular country?
- It is necessary to provide more fine-grained behavioral experiments related to agent cooperation to offer deeper insights. For example, examining how different cooperation modes, personalities, and thinking patterns of agents affect cooperation, rather than simply switching between different models for experiments.
-

**Strengths And Weaknesses:**

# Strengths
- Rigorous reproducibility of original claims, with clear alignment in results.
- Detailed reporting of computational costs and energy consumption enhances transparency.
- Well-structured experiments and comprehensive results facilitate interpretation.
# Weaknesses
- Lacking innovation, this paper only makes minor modifications based on GovSim, such as replacing the model, combining different models, or altering the environmental settings.
- Considering the influence of language on agent behavior, only Japanese was taken into account, and the experimental design was relatively simple. The experimental results lack persuasiveness.
- The experimental conclusions did not provide any additional insights; they still indicate that agents corresponding to more capable models exhibit stronger abilities.

---

> ### Author Response · Authors · 2025-04-18
>
> We thank the reviewer for the careful reading and for the positive, helpful comments. We have addressed all of them in detail below, and believe they led to a stronger paper.
>
> We respectfully disagree with the claim that our work lacks innovation or introduces only minor modifications to GovSim. While reproducibility studies naturally build on prior work, our extensions were chosen to explore meaningful new directions within computational limits. We go beyond model substitution by comparing fundamentally different architectures, including dense and MoE models, and show that open-weight alternatives can rival proprietary ones. We also introduce the MultiGov setup, a heterogeneous multi-agent system where stronger models influence weaker ones toward sustainable cooperation, suggesting a scalable path for cooperative systems.
>
> Moreover, we explore, in the “trash” scenario, how the framing of resource-sharing tasks affects agent behavior. Our results show that agents respond differently to equivalent environments depending on whether the resource is framed as desirable or not. Though limited to Japanese, our cross-lingual experiments were grounded in cultural theory. We also show that small models can improve significantly through interaction with stronger agents, enabling cost-efficient cooperation. We appreciate the reviewer’s suggestion on agent individuality and see it as an exciting future direction.
>
> ## Supplement the impact of different languages on agents
> We appreciate the suggestion and acknowledge the value of culturally specific backgrounds in evaluating cultural influences on LLMs. However, our goal was to isolate language effects without additional variables. Extending the study would require symmetric adaptations across languages and introduce new dimensions like geography and symbolism. Given our focus on reproducibility and resource limits, we did not pursue this path. We now note this in the Japanese Translation section and the Broader Impact and Ethical Considerations appendix.
>
> ## It is necessary to provide more fine-grained experiments ...
> We thank the reviewer for this suggestion. We fully agree that exploring more fine-grained behavioral dimensions, such as agent personalities or reasoning styles, would be a valuable direction for future work in multi-agent research. However, our study was designed as a reproducibility and extension effort, with a primary focus on preserving the original experimental setup proposed by Piatti et al. Introducing new behavioral variables would have shifted the focus away from reproducibility and risked confounding the validation of the original findings.
>
> Moreover, we note that the original paper already explored different modes of reasoning and cooperation through its use of varied environments (fishery, pasture, pollution), reasoning interventions (like the universalization principle), and scenario framing. These mechanisms were explicitly designed to capture how LLMs respond to different thinking patterns and ethical prompts, i.e., inputs that encourage the model to reflect on fairness. Part of our goal was to validate and test the robustness of those insights under new conditions (e.g., with new models, languages, and agent combinations).
>
> In our extensions, we aimed to go deeper rather than wider, introducing major changes such as heterogeneous-agent interactions, adversarial resource framing, without introducing entirely new behavioral abstractions. These choices were intentional so that we could isolate and evaluate how generalizable the original findings were across model architectures, linguistic settings, and social configurations. We see fine-grained behavioral modeling as a natural next step, and we appreciate the reviewer's encouragement in that direction. Therefore, we have added a paragraph in the conclusion to highlight this as a future research direction.
>
> ## Impact Concerns
> We fully agree that the ability of high-performing models to influence others, while promising for cooperative systems, also poses risks in adversarial settings, such as coordinated misinformation or manipulation. In response, we have added a dedicated section in the appendix titled Broader Impact and Ethical Considerations, where we explicitly discuss the dual-use implications of model influence in multi-agent contexts, and suggest concrete safeguards and future research directions to mitigate these risks.
>
> Regarding the cultural limitations of the Japanese experiment, we acknowledge the issues raised about translation quality and the absence of a culturally specific context. These limitations were shaped by resource constraints, but we agree they are important to address. We have now included a paragraph in the same appendix reflecting on the limitations of our cross-lingual design, and we emphasize this point again in the conclusion of the paper to signal its relevance for future work on cultural framing in LLM evaluation.

---

### Review · Reviewer_TW7z · 2025-04-08

**Summary Of Contributions:**

This study reproduces work Piatti et al. (2024), which introduced GovSim, a simulation framework designed to assess cooperative decision-making capabilities of large language models in resource-sharing scenarios. The authors replicate the part of the experiments of the original paper for Fishery problem, extended to cover more LLMs and Japanese-based simulation. The authors show that they can replicate the experiments and conclusions of Piatti  et al. The authors highlight a few novelties compared to Piatti  et al.:

Contributions are as follows:
1. Evaluation of New Models: The study tested additional LLMs, including DeepSeek-V3, GPT-4o-mini, and the Qwen family, showing DeepSeek-V3 matched GPT-4-turbo, while smaller models improved with universalization.
2. Introduction of a Japanese-Language Scenario: The authors introduced Japanese-language instructions in the Fishery scenario, finding no significant behavioral differences compared to English prompts.
3. Development of a New Inverse Trash Scenario: A novel scenario where agents eliminated harmful resources (trash) revealed that framing resources as undesirable can improve cooperation, enabling GPT-4o-mini to succeed.
4. Multi-Agent Heterogeneous Scenarios (MultiGov): High-performing models influenced low-performing models in mixed-agent setups, fostering sustainable behaviors through communication and cooperation.
5. Insights into Framing and Risk Aversion: The study showed agents behaved more cautiously in harmful-resource scenarios, highlighting the impact of framing on decision-making and cooperation.

**Audience:**

Yes

**Broader Impact Concerns:**

The study does not address the ethical implications of performing inverse experiment simulations, which involve simulating environmentally harmful issues.

**Claims And Evidence:**

Yes

**Requested Changes:**

1. Add a detailed motivation explanation for each introduced setting: (1)  Trash (2) MultiGov (3) Japanese.

2. Include multilingual simulation using heterogeneous agents framework, e.g one agent using English and another Japanese), emphasizing global collaboration.

**Strengths And Weaknesses:**

Strengths：

- The reproduction study is detailed and effectively supports the original claims, with results (e.g., Table 3) confirming that introducing universalization is crucial for achieving sustainability.
- Several novel settings are introduced to the original simulation, including heterogeneous multi-agent systems, inverse experiments, and language-based changes, extending the scope of the GovSim framework.
- Findings from the inverse experiments reveal how framing effects and risk aversion influence decision-making and cooperation, offering valuable insights with practical implications for real-world systems.

Weaknesses:

- The motivation behind several novel settings is not clearly stated, and their significance or differentiation from the original settings is insufficiently explained. For example, the inverse experiments appear similar to positive resource settings, as they seem to require the same cooperative abilities from agents. Further discussion on the importance of this setting in social science terms would clarify its value.
- The Japanese-language setting lacks depth in its practical insights, as it primarily tests models in a language they are weaker at. A more meaningful exploration could involve multilingual communication, such as scenarios where agents operate in different languages (e.g., one agent using English and another Japanese), emphasizing global collaboration.

---

> ### Author Response · Authors · 2025-04-18
>
> We would like to thank the reviewer for the careful reading of our manuscript and for the positive and helpful comments. We have replied to all of them, as described in detail below, and we believe that they have led to an improvement in the paper.
>
> ## Add a detailed motivation explanation for each introduced setting: (1) Trash (2) MultiGov (3) Japanese
>
> - We recognize the flaw pointed out by the reviewer in what comes to the motivation of the new settings. Given the importance of motivating or experiments, we have added to the main text a more detailed explanation for each of these three scenarios' purpose under section 2. Scope of Reproducibility.
>
> ## Include multilingual simulation using heterogeneous agents framework, e.g, one agent using English and another Japanese, emphasizing global collaboration
>
> - We thank the reviewer for the thoughtful suggestion to explore multilingual simulations within a heterogeneous agent framework. We agree that such a setup could provide valuable insights into the dynamics of global collaboration. However, in the current study, we intentionally varied one factor at a time, either model heterogeneity (in the MultiGov setting) or language (in the Japanese experiment), to ensure that the effects of each variable could be interpreted independently.
>
> - When introducing both dimensions simultaneously, different agents and languages would confound our ability to isolate the source of any behavioral changes. In practice, such settings also tend to cause models to revert to English during communication, even when prompted in other languages. This behavior is difficult to contain and would compromise experimental reliability.
>
> - Additionally, from a realist standpoint, practical cooperation in multilingual human societies typically depends on a shared common language. Simulating agents communicating in completely different languages without a shared medium would not reflect how multilingual collaboration typically unfolds in real-world settings.
>
> - Finally, the MultiGov configuration is already our most computationally demanding experiment. Extending it to a multilingual setup would exceed our current resource constraints.
>
> - That said, we appreciate the reviewer’s suggestion and fully agree that multilingual heterogeneous simulations are a valuable direction for future work. We have added this recommendation to our discussion of future research.
>
> # IMPACT CONCERNS
>
> ## The study does not address the ethical implications of performing inverse experiment simulations, which involve simulating environmentally harmful issues.
>
> - We thank the reviewer for raising this important concern and would like to clarify the intent of the inverse experiment. Rather than promoting or simulating environmentally harmful behavior, the inverse scenario is framed to test whether agents could cooperate to remove something harmful, specifically a shared pollutant or "trash" resource. This new setup allowed us to explore whether LLM agents respond differently when tasked with minimizing a shared problem or threat rather than maximizing a shared benefit.
>
> - We agree that care is needed when designing and interpreting such scenarios, especially in how they are related to real-world ethical concerns. We have now added a discussion to the  \textit{Broader Impact and Ethical Considerations} section in the appendix explaining motivation and clarifying that we don't promote destructive behavior through our experiment. A sentence in Section 3.2 of the main text now also acknowledges that and directs the reader to the appendix.

---

### Decision · Action_Editor_YGFJ · 2025-05-12

**Recommendation:** Reject

**Comment:**

The reviewers acknowledged the clarity of presentation, faithful replication of the original findings, and the effort to explore extensions such as heterogeneous multi-agent settings and inverse experiments.

After careful consideration, I have not recommended this submission for publication. The main concern is that the added extensions lack sufficient depth and are not framed in a way that clearly conveys their broader significance. Several of the new scenarios—such as the language variation—would benefit from stronger motivation and more careful analysis to distinguish their contributions from the original study. The whole work does not sufficiently establish its relevance or added value for the TMLR audience. I would be open to evaluating a major revision that strengthens the framing, expands the experiments, and clarifies the implications of the findings.

**Audience:**

Maybe — The topic may interest some readers, but the added contributions need stronger framing and clearer significance to fully engage the TMLR audience.

**Claims And Evidence:**

No — While the replication is acceptable, the added extensions lack sufficient depth.